# Identifying Molecular Modulators of the Vascular Invasion in Rectal Carcinoma: Role of *ADAMTS8* and Its Co-Dependent Genes

**DOI:** 10.3390/ijms26136261

**Published:** 2025-06-28

**Authors:** Bojana Kožik, Tarik Čorbo, Naris Pojskić, Ana Božović, Lidija Todorović, Ana Kolaković, Vesna Mandušić, Lejla Pojskić

**Affiliations:** 1Laboratory for Radiobiology and Molecular Genetics, Vinča Institute of Nuclear Sciences, National Institute of Republic of Serbia, University of Belgrade, Mike Petrovića Alasa 12–14, 11000 Belgrade, Serbia; bojana86@vin.bg.ac.rs (B.K.); anabozovic@vin.bg.ac.rs (A.B.); lidijat@vin.bg.ac.rs (L.T.); anakolakovic@vin.bg.ac.rs (A.K.); vvranic@vin.bg.ac.rs (V.M.); 2Laboratory for Bioinformatics and Biostatistics, Institute for Genetic Engineering and Biotechnology, University of Sarajevo, Zmaja od Bosne 8, 71000 Sarajevo, Bosnia and Herzegovina; tarik.corbo@ingeb.unsa.ba (T.Č.); naris.pojskic@ingeb.unsa.ba (N.P.); 3Laboratory for Human Genetics, Institute for Genetic Engineering and Biotechnology, University of Sarajevo, Zmaja od Bosne 8, 71000 Sarajevo, Bosnia and Herzegovina

**Keywords:** rectal carcinoma, vascular invasion, *ADAMTS8*, molecular docking

## Abstract

Rectal carcinoma (RC) represents approximately 30% of all colorectal carcinomas (CRC) and is considered a distinct clinical entity. Vascular invasion (VI) is recognized as an independent predictor of poor outcomes in RC. In this study, we applied bioinformatics methods to identify gene pathways most likely associated with VI in rectal carcinoma. As *ADAMTS8* showed statistically significant negative relations with the VI in RC patients, we further analyzed its top co-dependent genes—DNAL4, EVI2B, PPP1R35, PTGR3, RPL21, SOX4, and ZNF3—for the experimentally proven molecular modulators. We identified a total of 23 compounds from the Comparative Toxicogenomics Database based on previously reported data for all eight target genes. The search was expanded to include additional chemical agents by structure similarity using the PubChem database, which revealed 9661 additional compounds. These were subsequently used for molecular interaction analysis against target proteins co-expressed with, or associated with, *ADAMTS8* in RC with VI. Ultimately, we identified four high-affinity compounds—cyanoginosin LR, doxorubicin, benzo[a]pyrene, and dibenzo(a,e)pyrene—that interacted with all target proteins. These compounds show potential for further assessment of their role in modulating processes related to vascular invasion, which is a strong negative predictor of RC outcomes.

## 1. Introduction

Rectal carcinoma (RC) accounts for approximately 30% of all colorectal carcinomas (CRC) and is recognized as a distinct clinical entity with unique etiologies and treatment approaches [1]. Primary RC tumors progress to advanced stages through various mechanisms, some of which may serve as prognostic markers [2]. Extramural vascular invasion (EMVI) refers to the presence of cancer cells within blood vessels (both arterial and venous) located outside the muscularis propria of the rectum [3]. Venous invasion, a specific form of vascular invasion (VI), is increasingly recognized as a significant predictor of poor outcomes in colorectal cancer [2]. In clinical practice, venous invasion is routinely assessed and categorized into intramural and extramural types [4]. The anatomical location of the invaded vessel plays a crucial role in determining prognosis for patients with rectal cancer [2]. Studies have shown that vascular invasion is associated with poor prognosis, as confirmed by both pathological and MRI findings [5,6]. This association correlates with lower survival rates, higher risks of local and systemic recurrence, and increased mortality in RC patients [7].

The molecular mechanisms underlying venous invasion and VI in general [6] are poorly understood, with several potential factors contributing to the progression of the observed vascular changes. One key process is an epithelial-mesenchymal transition (EMT), which allows colorectal tumor cells to advance to more aggressive stages within the bowel wall, invade the blood and lymphatic vessels, and metastasize to local lymph nodes and distant sites [8]. Additionally, the degradation of the extracellular matrix (ECM) is crucial for tumor progression in this context [9]. Angiogenesis, the formation of new blood vessels, may also play a significant role in tumor dissemination via the vascular system [10]. Despite these insights, no reliable molecular biomarkers currently exist to accurately predict vascular invasion or to serve as potential therapeutic targets for VI-positive rectal tumors.

*ADAMTS8* and two other members of its gene family have previously been recognized as anti-angiogenic agents, drawing attention to them as potential tumor inhibitors [11]. The expression of *ADAMTS8* has been negatively correlated with tumor progression in several experimental models, including brain tumors, breast carcinoma, non–small cell lung carcinoma, head and neck squamous cell carcinoma, and pancreatic cancer, while in some other solid cancers, its role still remains unclear [11]. Several in vitro and in vivo studies have shown that upregulation of *ADAMTS8* could inhibit tumor growth in lung, breast, and colorectal cancers [12,13,14].

Vascular invasion is a known independent predictor of poor prognosis in rectal cancer, as evidenced by a higher risk of developing metastases and shorter disease-free survival (DFS) compared to VI-negative tumors. In vitro experiments have demonstrated that *ADAMTS8* overexpression can inhibit cell proliferation and migration. Additionally, markers of cell invasion and metastasis, such as beta-catenin, c-Myc, ERG, VEGFA, and CD31, were inversely reduced [14].

However, the molecular mechanisms of vascular invasion genesis in RC remain unclear. The objective of this study is to clarify the distinct molecular characterization of tumors with vascular invasion (VI) findings from that of VI-negative tumors, using established bibliographic data and bioinformatics tools. Given the critical role of vascular invasion in the metastatic cascade of colorectal cancer (CRC), we focused on genes that, according to existing literature [15,16,17,18,19,20,21,22,23,24,25,26,27,28,29,30,31,32,33,34,35,36], regulate various aspects of colorectal carcinogenesis. For that purpose, we explored the potential of external regulators of *ADAMTS8* expression as a basis for future directional treatment strategies.

## 2. Results

### 2.1. Patient Cohort and Dataset

The initial COADREAD dataset, accessed via the cBioPortal platform, consisted of 640 colorectal adenocarcinoma samples. After filtering for rectal adenocarcinoma, 169 patient samples remained. Among these, mRNA expression data were available for 92 samples, while vascular invasion status was determined in 78 samples. The selected cohort of 78 patients included 48.8% (38/78) male and 51.2% (40/78) female patients, with an average age of 64.4 ± 12.3 years (range 31–90). Among them, there were 17/78 VI-positive (21.8%) and 61/78 VI-negative (78.2%) rectal adenocarcinoma cases.

### 2.2. Association of mRNA Gene Expression and Methylation Levels of Selected Genes with Vascular Invasion

mRNA expression data for 41 preselected genes were available for all 78 samples. Among them, a relationship with vascular invasion was detected only for *ADAMTS8*, where lower mRNA expression levels were significantly associated with positive VI status (*p* = 0.039, tested by Mann–Whitney U test) (Figure 1A). No significant associations were observed for the remaining 40 genes (*p* ˃ 0.05 for all genes, Table 1), although a statistical trend was noted between lower *ADAMTS13* mRNA expression and VI-positive samples (*p* = 0.057), tested by Mann–Whitney U test).

We also downloaded methylation data for the *ADAMTS8* gene and observed a weak correlation between *ADAMTS8* mRNA expression and *ADAMTS8* methylation level in this cohort (*p* = 0.012, r = −0.263, Spearman’s test of correlation) (Figure 1B). Moreover, we observed a tendency toward an association between higher *ADAMTS8* methylation levels and vascular invasion (*p* = 0.087, tested by Mann–Whitney U test) (Figure 1C).

### 2.3. ADAMTS8 Signal Pathways, Interaction, and Clustering Analysis

To further investigate the role of *ADAMTS8* deregulated expression in rectal cancer pathogenesis, we analyzed correlations between *ADAMTS8* mRNA expression and the expression of the other 40 preselected genes in our cohort (Appendix A). A statistically significant moderate positive correlation was observed with ZEB1 (*p* < 0.001, r = 0.501). Weak but significant correlations were also identified with ADAMTS1 (*p* < 0.001, r = 0.467), VCAM1 (*p* < 0.001, r = 0.367), ZEB2 (*p* = 0.002, r = 0.324), COL4A2 (*p* = 0.002, r = 0.323), and TIMP2 (*p* = 0.004, r = 0.296) (Figure 2). Additionally, very weak but statistically significant correlations were noted in cases of VIM, TGFβ1, TGFβ2, MMP2, MMP3, and MMP10 (*p* < 0.05 in all tests) (Appendix A).

To further examine potential interactions between *ADAMTS8* and its positively correlated genes, we performed protein-protein network and cluster analysis using the STRING online tool. The results demonstrated that queried proteins formed a significantly enriched interaction network (PPI enrichment *p*-value: <1.0 × 10^−16^), which was organized into tree functional clusters (Figure 3). Biological Process (Gene Ontology) and Reactome Pathways enrichment analysis revealed that most of the queried proteins are involved in extracellular matrix organization processes (Figure 4), while Disease-gene Associations (DISEASES) enrichment associated the queried proteins only with arterial disease. Interestingly, protein-protein interaction and cluster analysis revealed that the ADAMTS8 protein did not belong to any cluster and showed no direct interactions with the other queried proteins.

### 2.4. Prediction of ADAMTS8 Co-Dependent Gene Targets and Their Association with Vascular Invasion

To expand the list of potential *ADAMTS8* co-expressed genes and/or functional modulators, we used the Predictability tool available on The DepMap portal website. From the CRISPR and shRNA lists of the top co-dependent *ADAMTS8* genes, we selected the top seven genes (DNAL4, EVI2B, PPP1R35, PTGR3, RPL21, SOX4, and ZNF3), that negatively correlate (according to the Pearson correlation parameter) with *ADAMTS8* expression. Next, we evaluated the association of mRNA expression of all seven targets in our selected cohort of patients with vascular invasion (Appendix A).

Mann–Whitney U tests showed significant relations between lower levels of DNAL4 and PTGR3 mRNA expression and vascular invasion (*p* = 0.037 and *p* = 0.022, respectively) (Figure 5). On the contrary, a significant relationship between higher levels of RPL21 mRNA expression and vascular invasion was observed (*p* = 0.023). Moreover, in our cohort of 78 patients, we noted statistically significant weak correlations of *ADAMTS8* mRNA expression with EVI2B mRNA expression (*p* = 0.002, r = 0.352), SOX4 mRNA expression (*p* = 0.011, r = −0.288) and RPL21 mRNA expression (*p* = 0.001, r = −0.384) (Figure 5, Appendix A).

### 2.5. Molecular Docking Analysis

We compiled a large database of compounds based on molecular similarity indices and employed target protein-ligand docking, a computational technique commonly used to predict the inhibitory effects of compounds on specific proteins, to identify the strongest pathway regulators shared between target proteins. Molecular docking studies involving 9684 compounds against multiple target proteins, including ADAMTS8, DNAL4, EVI2B, PPP1R35, PTGR3, RPL21, SOX4, and ZNF3, revealed strong binding affinities between these compounds and their respective protein targets. Given the extensive number of available compounds, a subset of five compounds from the initial 23 (Table 2), along with 10 additional compounds from the 9661 similar structures (Table 3), was selected per target protein based on binding affinity for further analysis.

This selection allowed for a detailed investigation of molecular interactions and the nature of the formed bonds. Visualization using Discovery Studio Visualizer provided comprehensive insights into these interactions, revealing an intricate network of bonds between the selected compounds and key amino acid residues of the target proteins.

Among the initially tested 23 compounds, benzo[a]pyrene, cyanoginosin LR, and doxorubicin exhibited notable pan-target activity, forming stable interactions with all investigated proteins. Their consistent binding profiles across the entire target set suggest a broad-spectrum affinity, positioning them as promising pan-target ligand candidates for further validation. Detailed information on the binding affinity and the molecular interactions, including the types and lengths of bonds formed in these top-ranked docked complexes, is provided in Appendix A.

#### 2.5.1. Benzo[a]pyrene

Among the three detected pan-target ligands, benzo[a]pyrene exhibited strong binding affinities across all eight protein targets by forming a diverse array of non-covalent interactions, primarily involving pi–alkyl, pi–pi stacked, pi–pi T-shaped, pi–sigma, and hydrogen bonds (Appendix A). Among these, pi–pi stacked interactions, frequently established with aromatic residues such as Phe, Trp, and Tyr, were particularly notable due to their planarity and stacking geometry, which promote both binding strength and molecular complementarity. This was exemplified by interactions with Phe396 in ADAMTS8, Trp225 in EVI2B, and Trp125 in RPL21. Additionally, pi–alkyl interactions with hydrophobic residues such as Leu, Ile, and Arg contributed to the stabilization of the ligand within nonpolar binding pockets. The presence of pi–pi T-shaped interactions, such as those with His363 in ADAMTS8 and Tyr90 in DNAL4, further enhanced the binding by introducing flexible aromatic contacts. Notably, a hydrogen bond with Thr150 in PTGR3 provided critical specificity and anchoring potential, indicative of strong local complementarity. Overall, the most extensive and potentially strongest interaction profile was observed with EVI2B, where benzo[a]pyrene formed multiple pi–alkyl and an exceptional number of pi–pi stacked interactions with Trp225, suggesting a highly stable and favorable binding configuration.

#### 2.5.2. Cyanoginosin LR

Cyanoginosin LR demonstrated even greater interaction diversity, engaging in a broad spectrum of hydrogen bonds, pi–alkyl, alkyl, pi–pi stacked, pi–pi T-shaped, and pi–sigma interactions with all eight protein targets (Appendix A). Hydrogen bonding was the dominant interaction type, particularly in complexes with ADAMTS8, DNAL4, and ZNF3, where numerous interactions were formed with polar and charged residues such as Arg780, His70, and Tyr76, many of which fell within a strong bonding distance range (<3.5 Å). These bonds significantly contributed to the directionality, specificity, and stability of the ligand–protein complexes. In addition, pi–alkyl and alkyl interactions with hydrophobic residues including Leu, Val, Ile, and Phe played a key role in enhancing hydrophobic complementarity, particularly in EVI2B, PTGR3, and ZNF3. The presence of pi–pi stacking (e.g., Phe472 in SOX4 and Phe89 in PTGR3) and pi–pi T-shaped interactions (e.g., Phe11 in EVI2B) further stabilized the ligand orientation within aromatic-rich environments. Notably, ADAMTS8 emerged as the most favorable binder for cyanoginosin LR, forming over 15 hydrogen bonds with residues such as Arg780, Gly680, and Cys666, in addition to hydrophobic contacts, suggesting a deeply anchored and highly stable binding conformation.

#### 2.5.3. Doxorubicin

Doxorubicin exhibited high binding affinities with all eight proteins, forming a rich network of hydrogen bonds, pi–alkyl, pi–pi T-shaped, pi–cation, pi–anion, pi–sigma, and alkyl interactions (Appendix A). Among these, hydrogen bonds were the most prevalent and essential, contributing significantly to complex stability and binding specificity, especially through strong interactions with polar and charged residues such as Gln183, Arg210, Gln660, and Cys661 in ADAMTS8, as well as Glu75, Arg229, Asn234, and Tyr152 across other targets. Pi–alkyl interactions with hydrophobic and aromatic residues such as His182, Val662, Ala54, and Trp238 contributed additional hydrophobic stabilization, enhancing the ligand’s fit within protein pockets. Notably, pi–pi T-shaped interactions with aromatic residues (e.g., His182 in ADAMTS8 and Phe215 in PPP1R35) and pi–sigma contacts (e.g., with Leu231 in PPP1R35 and Lys156 in PTGR3) reinforced binding orientation through multipoint aromatic stacking. The presence of pi–cation and pi–anion interactions with Arg210 and Asp185, respectively, added electrostatic complementarity, enhancing the total binding energy. Among all proteins, ADAMTS8 exhibited the most diverse and abundant interaction pattern, featuring hydrogen bonds, pi–pi interactions, alkyl contacts, and electrostatic interactions, suggesting a highly favorable binding site and potentially the strongest overall complex stability with doxorubicin.

From the 9661 compounds obtained through 3D similarity screening, only dibenzo(a,e)pyrene demonstrated consistent binding across all eight protein targets. This compound established stable interactions with each protein, highlighting its unique profile as the sole candidate with pan-target potential among the screened molecules. Comprehensive data on the binding affinity and the molecular interactions, including bond types and bond lengths for all top-performing compounds, are provided in Appendix A.

#### 2.5.4. Dibenzo(a,e)pyrene

Dibenzo(a,e)pyrene exhibited a broad range of non-covalent interactions across all eight target proteins, marked by pi–pi stacking, pi–alkyl, pi–sigma, pi–anion, pi–cation, and hydrogen bonding (Appendix A). Among these, pi–pi stacked interactions played a dominant role, particularly evident in complexes with EVI2B (via Trp225) and PPP1R35 (via Phe215), where extensive aromatic stacking contributed significantly to complex stabilization and binding orientation. Pi–alkyl interactions were frequent and involved residues such as Leu332, Leu395, Leu493, Ile222, and Lys226, enhancing the hydrophobic character of binding pockets and favoring ligand accommodation. Pi–sigma interactions, though less frequent, were observed with residues like Thr176 and Gln232, adding geometric complementarity. Notably, pi–anion bonds with negatively charged residues (e.g., Glu353 in ADAMTS8 and Asp38, Asp103 in RPL21) provided electrostatic stabilization, while pi–cation interactions (e.g., with Arg77 in ZNF3) added further polar complementarity. A single hydrogen bond with Ile39 in RPL21 (3.04 Å) reinforced local stabilization. Among all targets, EVI2B and ZNF3 exhibited the highest density and diversity of interactions, particularly through extensive pi–pi stacking, suggesting these proteins may have the most favorable binding sites for dibenzo(a,e)pyrene and potentially the strongest overall binding affinity.

## 3. Discussion

This study was conducted to identify genes and pathways specific to a VI-positive rectal adenocarcinoma phenotype. We examined the expression data of 41 preselected, literature-supported gene candidates involved in regulating EMT, ECM, and angiogenesis during colorectal carcinogenesis and their potential association with VI using a multi-omics RC dataset, with the ultimate goal of validating potential molecular targets for repurposing and targeted treatment of VI-positive rectal cancer patients.

### 3.1. ADAMTS8 as a Critical Target for VI Modulation

To better understand the tumor microenvironment of vascular invasion (VI)-positive rectal tumors, which consist of tumor cells and surrounding stroma, we focused on the expression profiles of genes involved in extracellular matrix (ECM) remodeling. Among the 41 gene candidates tested, we found a significant association between lower mRNA expression of the *ADAMTS8* gene and vascular invasion (*p* = 0.039). To the best of our knowledge, this is the first report highlighting the significant role of *ADAMTS8* in regulating vascular invasion in rectal carcinoma. The ADAMTS family members are involved in extracellular matrix remodeling, which is crucial for cancer cell invasion and metastasis [37]. Specifically, ADAMTS8 has been identified as a potential tumor suppressor in colorectal cancer [13] and is frequently silenced in carcinomas, including CRC, due to promoter methylation [11]. This silencing is significant because *ADAMTS8* plays a role in inhibiting tumor cell growth and motility by antagonizing the EGFR-MEK-ERK signaling pathway [11], which is crucial for cancer cell proliferation and survival. Results of our study indicate that promoter methylation of the *ADAMTS8* gene acts as a mechanism for epigenetic gene silencing. In this cohort, we observed a significantly weak correlation between *ADAMTS8* mRNA expression and its methylation level (*p* = 0.012, r = −0.263). Additionally, there was a trend suggesting that higher levels of *ADAMTS8* methylation may be associated with vascular invasion (*p* = 0.087).

According to the literature, ADAMTS8 also exhibits anti-angiogenic properties, as its thrombospondin motifs interact with the CD36 receptor on endothelial cells [38]. Our study results showed a weak but significant positive correlation between *ADAMTS8* and *VCAM1* (vascular cell adhesion protein 1) expression. It is well established that VCAM1 promotes transendothelial migration in colorectal cancer [17], which could be a mechanism behind vascular invasion. Kokelaar (2021) [39], reported that the expression of *MMP2* (matrix metallopeptidase 2) is significantly associated with extramural venous invasion (EMVI), a specific type of vascular invasion in rectal cancer. However, we did not observe such a correlation between MMP2 and vascular invasion in our study. That said, we found a very weak but statistically significant positive correlation between *MMP2* and *ADAMTS8* mRNA expression. Additionally, there was a near-significant positive correlation between *ADAMTS8* mRNA expression and *HIF1A* (hypoxia-inducible factor 1 subunit alpha) mRNA expression in our study (*p* = 0.058, r = 0.199). This is notable since HIF1A is known to promote abnormal angiogenesis in colorectal cancer [40]. Furthermore, a recent study indicated that *ADAMTS8* inhibits the progression of lung cancer by suppressing the key angiogenesis factor, VEGFA (vascular endothelial growth factor A) [12]. In our study, we also observed a tendency for *ADAMTS8* mRNA expression to negatively correlate with *VEGFA* mRNA expression (*p* = 0.074, r = −0.187). However, VEGFA expression did not show a direct relationship with vascular invasion. These results indirectly suggest the important role of aberrant *ADAMTS8* expression in regulating vascular invasion in the pathogenesis of rectal cancer.

Beyond CRC, ADAMTS8 is also recognized for its tumor-suppressive and anti-angiogenic roles across various cancer types. In breast cancer, it inhibits cell proliferation, migration, and invasion, with overexpression linked to improved survival outcomes and acting as an independent prognostic factor by regulating the EGFR/Akt pathway [41]. In lung cancer, ADAMTS8 enhances survival, especially in patients with wild-type EGFR or low PD-L1 expression, by promoting anti-cancer NKT cells and reducing immunosuppressive T cells [42]. Its downregulation in esophageal squamous cell carcinoma correlates with advanced stages and lymph node metastasis, while overexpression promotes apoptosis [43]. In hepatocellular carcinoma, ADAMTS8 reduces cell proliferation and metastasis via the ERK pathway, with low expression indicating poor prognosis [44]. *ADAMTS8* is significantly downregulated in brain tumors, including high-grade gliomas, and plays a role in inhibiting angiogenesis by affecting stromal-epithelial interactions and matrix composition [45]. Similar to the results of our study, in gastric cancer, *ADAMTS8* downregulation, often linked to DNA hypermethylation, correlates with increased invasion and metastasis, as its expression is inversely related to invasive depth and lymph node metastasis [46]. Nonetheless, the exact molecular mechanisms behind *ADAMTS8*-mediated progression of vascular invasion in RC require further investigation in a larger cohort of patients with vascular invasion-positive rectal cancer.

### 3.2. Common Targets of Modulators Associated with VI

Restoration of *ADAMTS8* expression has been shown to suppress tumor cell clonogenicity and induce apoptosis, highlighting its potential as a therapeutic target and biomarker in CRC [11]. The methodology further applied in this study was employed for the identification of bioactive compounds from existing chemical libraries [47]. To expand the list of possible VI regulators, we also included the top 7 *ADAMTS8* co-dependent genes (*DNAL4*, *EVI2B*, *PPP1R35*, *PTGR3*, *RPL21*, *SOX4*, *ZNF3*) in our drug screening. According to recent literature data, some of these genes already demonstrate oncogenic roles in CRC progression, especially the invasion processes, making them potential targets for anti-cancer therapy. For example, *EVI2B* (Ecotropic Viral Integration Site 2B) represents a lymphocyte-specific attractor gene which possesses a predictive role in CRC [48]. The *SOX4* (SRY-BOX Transcription Factor 4) gene expression is upregulated in CRC, which promotes tumor cell migration and invasion [49], while Zhu et al. (2023) reported that high expression of RPL21 (Ribosomal Protein L21) induces invasion and metastasis in CRC [50]. The *PTGR3* gene (phosphatase of regenerating liver-3, or PRL-3) correlates significantly with tumor progression and patient outcomes in colorectal cancer (CRC) [51]. High PRL-3 levels are linked to more aggressive tumor characteristics, including increased invasion and liver metastasis [52]. ZNF3, a zinc-finger transcription factor, is highly expressed in CRC and associated with increased cell proliferation, migration, and invasion. It may promote tumorigenesis by regulating markers such as MMP1 and TWIST, involved in epithelial-mesenchymal transition (EMT) [53]. While the mechanisms of *PPP1R35* in cancer progression are unclear, its expression suggests a role in breast cancer metastasis [54]. Similarly, *PPP1R14A* is linked to poor prognosis across cancers, including rectal carcinoma, and correlates with immune cell infiltration [55]. Finally, the *DNLA4* gene (DNA Ligase 4 or Lig4) is crucial for DNA repair through non-homologous end joining (NHEJ). However, genetic variations in Lig4 do not significantly affect CRC risk [56]. Since the expression of these genes demonstrated a negative correlation with *ADAMTS8* expression according to the DeepMap analysis, we consider all of them as possible inhibitors of *ADAMTS8* expression, and therefore suitable targets for molecular docking analysis.

In our iterative toxicogenomic screening, we identified four compounds that strongly interact with multiple candidate gene products, making them potential pathway regulators. Cyanoginosin LR belongs to the class of microcystins that are the most prevalent toxins produced by freshwater cyanobacteria and demonstrate antitumor properties through both immune system activation and direct cell toxicity in preclinical studies. Li et al. (2024) report that intraperitoneal administration of microcystin-LR together with oxaliplatin in a mouse colorectal cancer model produces significant tumor growth inhibition [57]. In that setting, treatment coincides with marked immune activation—documented by an increase in CD86^+^ M1 macrophages, maturation of CD11c^+^ dendritic cells, and a rise in CD8^+^ and PD-1^+^ T-cells alongside elevated interferon-γ and granzyme B levels. In parallel, Monks et al. (2007) [58] demonstrate that HeLa cells engineered to express the transport proteins OATP1B1 and OATP1B3 became over 1000-fold more sensitive to microcystin-LR, with certain analogs inducing rapid cell death at half-maximal inhibitory concentrations below 1 nmol/L. These findings support the conclusion that microcystin-LR—akin to cyanoginosin LR as posed in the research question—can function as an antitumor agent through both immune modulation and transporter-mediated direct cytotoxicity.

Doxorubicin plays a significant role in colorectal cancer treatment, but its effectiveness varies. Doxorubicin-induced cell death in colorectal cancer cells was observed with nanoencapsulation and a targeted delivery approach [59]. The miR-223/FBXW7 pathway regulates doxorubicin sensitivity through epithelial-mesenchymal transition in colorectal cancer cells, with miR-223 overexpression decreasing sensitivity [60]. To improve targeting and reduce side effects, hyaluronic acid-doxorubicin nanoparticles have been developed. These nanoparticles accumulate in the intestines, preserve mucosal epithelial integrity, and decrease the expression of apoptotic and inflammatory markers in a mouse model of chemically induced colon cancer [61].

On the contrary, some studies show that doxorubicin, a widely used chemotherapeutic agent, promotes cancer cell migration and invasion through various mechanisms. In breast cancer cells, doxorubicin enhances migration and invasion by activating the RhoA/MLC pathway [62]. Doxorubicin contributes to thrombus formation and vascular injury by enhancing platelet functions, including aggregation, degranulation, and adhesion to endothelial cells [63]. Furthermore, doxorubicin-induced vascular remodeling involves the dysregulation of various microRNAs, which affect signaling pathways crucial for vascular homeostasis and angiogenesis [64]. These findings highlight the complex effects of doxorubicin on cancer progression and vascular health, emphasizing the need for targeted strategies to mitigate its adverse effects while maintaining its therapeutic efficacy.

Dibenzo[a,e]pyrene (DBP) and benzo[a]pyrene (B[a]P) are potent polycyclic aromatic hydrocarbon (PAH) carcinogens. Polycyclic aromatic hydrocarbons are widespread environmental contaminants. Benzo[a]pyrene (B[a]P), a well-studied example, exerts toxicity through oxidative metabolism, producing reactive intermediates that alter cell structure and function. Recent evidence suggests B[a]P toxicity also disrupts cellular signaling pathways regulating growth and differentiation [65]. DBP is approximately 100 times more carcinogenic than B[a]P, primarily due to its molecular structure and inefficient DNA adduct repair [66]. Both compounds require metabolic activation by cytochrome P450 enzymes, particularly CYP1B1, to form DNA-reactive metabolites [67]. B[a]P is commonly found in grilled foods, air, and water, while DBP exposure sources are less clear [68]. In utero exposure to DBP in mice leads to aggressive T-cell lymphoblastic lymphoma in offspring, with the aryl hydrocarbon receptor (AHR) playing a role in susceptibility [69]. These PAHs cause various cancers by forming DNA adducts, inducing mutations, and altering gene expression. The carcinogenic potency of PAHs is influenced by their molecular structure, metabolic activation, DNA binding, and repair efficiency [66].

Benzo[a]pyrene (B[a]P) is a polycyclic aromatic hydrocarbon formed during incomplete combustion of organic materials [68]. It is classified as a Group I carcinogen, exhibiting mutagenic and carcinogenic properties [70]. B[a]P induces oxidative stress by generating reactive oxygen species and disrupting antioxidant mechanisms [70]. Its toxicity is mediated through metabolic activation, leading to DNA adduct formation and cellular signaling pathway disruption [65,68]. B[a]P exposure, often through cigarette smoke and grilled foods, has been linked to various cancers, atherosclerosis, and inflammatory diseases [71]. It activates the aryl hydrocarbon receptor (AHR), influencing xenobiotic metabolism and immune responses [71]. Several compounds, including vitamins and plant-derived substances, have shown potential in reducing B[a]P-induced oxidative stress [70].

## 4. Materials and Methods

### 4.1. Gene Candidate Selection

The main selection criterion was the involvement in vascular invasion, i.e., epithelial-mesenchymal transition, remodeling of the extracellular matrix, and angiogenesis. Most of these genes are also included in standard panels for profiling human tumor metastasis. The 41 genes selected for this study encode several classes of protein factors including cell adhesion proteins (*CDH1*, *OCLN*, *VCAM1*), ECM components (*COL4A2*, *VCAN*, *VTN*, *ECM1*), ECM modulators (*MMP2*, *MMP3*, *MMP7*, *MMP8*, *MMP9*, *MMP10*, *MMP11*, *MMP13*, *TIMP1*, *TIMP2*, *TIMP3*, *TIMP4*, *ADAMTS1*, *ADAMTS8*, *ADAMTS13*, *LOXL2*), cell cycle regulators (CDKN2A), apoptosis regulators (MDM2), angiogenesis factors (*VEGFA*, *VEGFB*, *VEGFC*), transcription factors (*ZEB1*, *ZEB2*, *TWIST1*, *SNAI1*, *SNAI2*, *SNAI3*, *HIF1α*), cytoskeleton components (*VIM*), and multifunctional receptors and regulators (*NOTCH-1*, *TGFβ1*, *TGFβ2*, *TGFβ3*, *APC*) related to CRC tumor metastasis.

### 4.2. cBioPortal Data Selection and Processing

To perform integrative biostatistics analysis, we searched for suitable RC datasets at the cBioPortal platform (https://www.cbioportal.org/ accessed on 18 August 2024), a publicly available repository of genomics and transcriptomics data for various tumor types. For this study, we chose COADREAD expression dataset including clinical metadata for colorectal cancer patients, which was derived from the TCGA Firehose Legacy large-scale study (https://gdac.broadinstitute.org/ accessed on 18 August 2024). The criteria for the selection of samples for biostatistics analysis were primary rectum tumor location, absence of other malignant diseases, and the pathologically determined status of vascular invasion (which includes both extramural and intramural venous invasion) as a separate clinical parameter from lymphatic invasion. Data filtering ensured the preservation of the clinical annotation quality for rectal carcinoma. After grouping samples according to the criteria, 41 previously selected genes were queried to obtain their mRNA expression levels and methylation patterns. The mRNA expression levels were available from the Illumina next-generation sequencing data and presented as the RNASeq V2 RSEM dataset, while DNA methylation data was derived from the HumanMethylation450 (HM450) Infinium array. Downloaded clinical and molecular data for selected samples were examined using a classical statistical approach.

### 4.3. Statistical Analysis

All statistical analyses were performed using MedCalc^®^ Statistical Software version 22 (MedCalc Software Ltd., Ostend, Belgium; https://www.medcalc.org accessed on 18th August 2024), while *p* value < 0.05 was considered to indicate statistical significance. For graphical presentation of results, the SigmaPlot 14.0 licensed statistical software package was used. The Kolomogorov-Smirnov test was applied to test for a normal distribution. The Mann–Whitney test was used to compare mRNA expression and methylation level of selected genes grouped according to the vascular invasion status, while correlations between mRNA expression and methylation level were estimated by the Spearman’s rank correlation test.

### 4.4. Enrichment Analysis for Vascular Invasion-Related Gene Product

To investigate the protein interaction landscape of genes associated with vascular invasion (VI) in rectal carcinoma, we performed a protein-protein interaction (PPI) network analysis using the STRING database (v11.5) (https://string-db.org accessed on 18 August 2024). The query list consisted of the *ADAMTS8* gene and genes whose expression correlated with *ADAMTS8* expression in our cohort of RC patients. STRING query parameters were set as follows: 1. Input proteins (gene symbols): *ADAMTS8*, *ZEB1*, *ZEB2*, *VIM*, *TGFB1*, *TGFB2*, *MMP2*, *MMP3*, *MMP10*, *TIMP2*, *COL4A2*, *VCAM1*, and *ADAMTS1*; 2. Organism: Homo sapiens (NCBI Taxonomy ID: 9606); 3. Minimum required interaction score: 0.400 (medium confidence); 4. Network edges: Evidence-based (not prediction-based), showing only evidence of functional associations; 5. Active interaction sources: Text mining, experiments, databases, co-expression, neighborhood, gene fusion, and co-occurrence were all enabled; and 6. Max number of interactors: 1st shell (direct interactors): none/query proteins only, and 2nd shell (indirect interactors): 0 (disabled). The resulting network was exported and visualized using the STRING graphical interface. Confidence scores of individual interactions were used to identify the most strongly interconnected proteins.

To identify genes functionally associated with our primary target gene, *ADAMTS8*, we utilized the DepMap (Dependency Map) portal (https://depmap.org/portal/ accessed on 10 October 2024), a comprehensive platform that integrates CRISPR and RNAi screening data across a broad range of cancer cell lines. The analysis was conducted using the “Predictability” tool after *ADAMTS8* was queried across the CRISPR (DepMap Public 24Q4+ Score, Chronos) and RNAi (Achilles + DRIVE + Marcotte, DEMETER2) datasets. Based on the results of Core Omics predictive models, we selected the top seven mRNA gene expressions with a negative correlation with *ADAMTS8* expression, as determined by Pearson correlation coefficients. These genes were prioritized as potential *ADAMTS8* downregulators and included in downstream protein structure modeling and molecular docking analysis due to their potential role in vascular invasion mechanisms.

### 4.5. Molecular Docking Analysis Workflow

#### 4.5.1. 3D Structures Selection

The three-dimensional crystal structures of the target proteins were obtained from UniProt and the RCSB Protein Data Bank (PDB) in PDB format (UniProt Consortium, 2025) [72,73]. The selected proteins include A disintegrin and metalloproteinase with thrombospondin motifs 8 (UniProt ID: Q9UP79), dynein axonemal light chain 4 (UniProt ID: O96015), protein EVI2B (UniProt ID: P34910), protein phosphatase 1 regulatory subunit 35 (UniProt ID: Q8TAP8), prostaglandin reductase 3 (PDB ID: 7ZEJ), large ribosomal subunit protein eL21 (PDB ID: 8FLD), transcription factor SOX-4 (UniProt ID: Q06945), and zinc-finger protein 3 (UniProt ID: P17036). Since the experimental structures were initially obtained in complex with other molecules, all non-target components were carefully removed to ensure a focused structural analysis.

#### 4.5.2. Protein Structure Preparation

Protein preparation was conducted using AutoDock Tools 1.5.6 [74], involving the removal of water molecules, addition of polar hydrogen atoms, and assignment of Kollman charges. The processed protein structures were then converted from PDB to PDBQT format, and grid boxes were defined to ensure proper docking site localization. The grid box dimensions and center coordinates were as follows: ADAMTS8 (116 × 66 × 96 Å, centered at −3.254, −4.668, −0.129), DNAL4 (56 × 36 × 40 Å, centered at −15.542, 1.767, −2.282), EVI2B (126 × 126 × 126 Å, centered at −13.744, 4.242, −4.095), PPP1R35 (118 × 70 × 118 Å, centered at −6.307, −2.954, 4.887), PTGR3 (72 × 60 × 88 Å, centered at −6.291, −4.262, 21.235), RPL21 (48 × 66 × 78 Å, centered at 308.383, 220.423, 297.309), SOX4 (126 × 118 × 126 Å, centered at −6.879, 12.299, −1.669), and ZNF3 (118 × 94 × 86 Å, centered at 7.325, 17.672, −7.343). For all docking simulations, the grid spacing was set to 1.0 Å.

#### 4.5.3. Candidate Compounds Selection and Preparation

Initially, 23 compounds were selected from the Comparative Toxicogenomics Database based on their reported interactions with ADAMTS8 [75]. Following this selection, a similarity search was conducted in the PubChem database, utilizing the 3D shape and structural features of these compounds as reference points. This search yielded a total of 9661 additional compounds, which were subsequently downloaded for further analysis [76]. As described by Čorbo et al. (2023) [77], the compounds were initially obtained in SDF format and then prepared for docking by converting them into PDBQT format using OpenBabel 3.1.1 with default parameters. During this preparation process, several modifications were applied, including the addition of charges and hydrogen atoms, assignment of atom types, conversion of bond types, and definition of the root. Notably, OpenBabel’s default settings generate 3D molecular structures in a neutral state, without considering potential ionization states. This property was preserved in the final PDBQT format of the ligands used in this study [78].

#### 4.5.4. Molecular Interactions Simulations

Molecular docking simulations were performed using AutoDockVina 1.1.2, employing a blind docking approach with default parameters for energy range and exhaustiveness [79]. Following docking, the interactions between the receptor proteins and the selected compounds were visualized and analyzed using PyMOL 3.1 and BIOVIA Discovery Studio Visualizer.

## 5. Conclusions

Our study identifies *ADAMTS8* as potentially associated with a histopathological subtype of rectal cancer. In silico analysis of 9684 compounds, selected from the Comparative Toxicogenomics Database along with their structural analogs, was conducted to evaluate their interactions with *ADAMTS8* and its seven, top co-dependent genes (*DNAL4*, *EVI2B*, *PPP1R35*, *PTGR3*, *RPL21*, *SOX4*, *ZNF3*). The screening identified cyanoginosin LR, doxorubicin, benzo[a]pyrene, and dibenzo(a,e)pyrene as the most consistent and strongly interacting candidate compounds, demonstrating strong binding affinities and stable interactions with all eight target proteins. Our findings suggest that these compounds may be directly engaged with venous invasion in rectal cancers, a key histopathological change associated with poor prognosis in rectal carcinoma. However, further experimental validation, including in vivo studies, as well as clinical trials, is necessary to confirm their true role and potential modulating effect in the onset and progression of rectal cancers with vascular invasion, as well as their therapeutic potential.

## Figures and Tables

**Figure 1 ijms-26-06261-f001:**
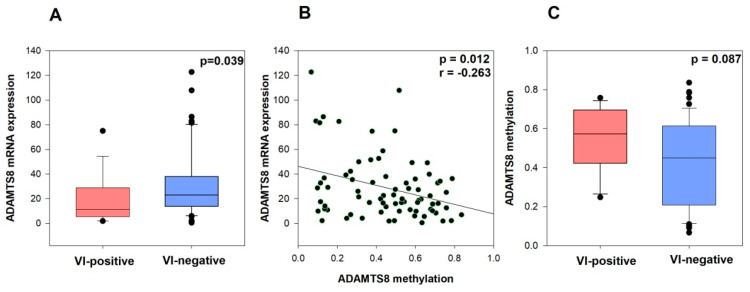
*ADAMTS8* mRNA expression in VI-negative and VI-positive groups (**A**), Correlation between *ADAMTS8* mRNA expression and *ADAMTS8* methylation (**B**), *ADAMTS8* methylation in VI-negative and VI-positive groups and vascular invasion in RC dataset (**C**).

**Figure 2 ijms-26-06261-f002:**
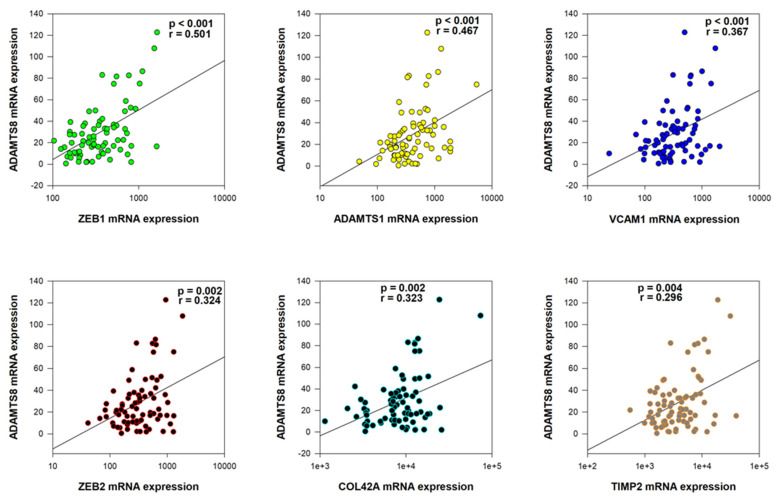
The significantly positive moderate/weak correlations with *ADAMTS8* mRNA expression.

**Figure 3 ijms-26-06261-f003:**
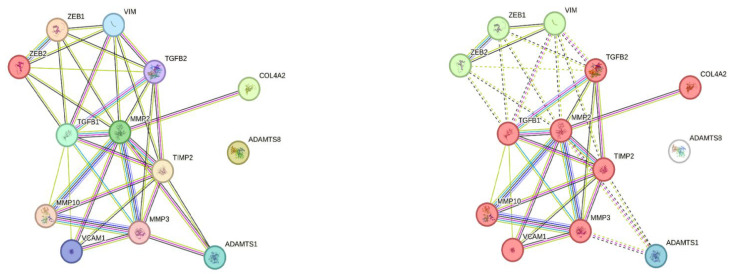
ADAMTS8 protein-protein interaction network in STRING platform. According to protein-protein network analysis, only the ADAMTS8 protein displays no physical or functional interaction with other proteins whose expression correlates with *ADAMTS8* expression in our cohort of RC patients (**left diagram**); Clustering analysis separated queried proteins in three clusters: Cluster 1 (red)—Activation of Matrix Metalloproteinases, Cluster 2 (green)—C-terminal binding protein, and Hypoxia-mediated EMT and stemness, and Cluster 3—ADAMTS1 (blue), while *ADAMTS8* does not belong to any of three clusters (**right diagram**).

**Figure 4 ijms-26-06261-f004:**
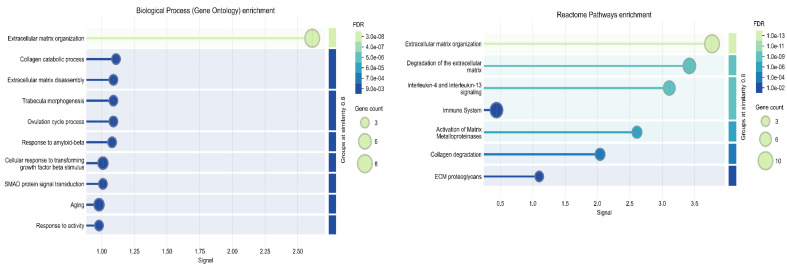
Biological Process (Gene Ontology) enrichment (**left diagram**) and Reactome Pathways enrichment (**right diagram**) analysis of ADAMTS8 and its correlated proteins in STRING platform.

**Figure 5 ijms-26-06261-f005:**
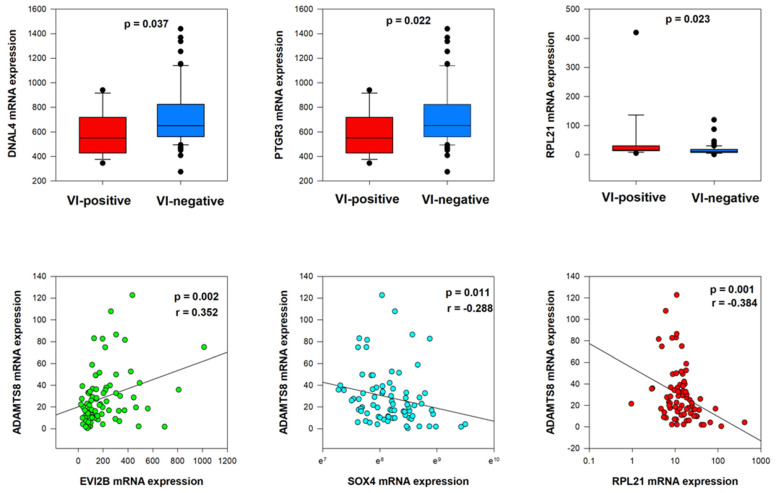
Co-dependent genes’ expression (DNAL4, PTGR3, and RPL21) in VI-positive and VI-negative groups and correlation with *ADAMTS8* mRNA expression.

**Table 1 ijms-26-06261-t001:** Association between 41 preselected genes mRNA expression and vascular invasion in RC.

Gene	Vascular Invasion (N)	Mean Rank	Sum of Ranks	Z	*p*
*APC*	Negative (61) Positive (17)	40.23 36.88	2454.00 627.00	−0.539	0.590
*CDH1*	Negative (61) Positive (17)	38.52 43.00	2350.00 731.00	−0.720	0.471
*CDKN2A*	Negative (61) Positive (17)	39.74 38.65	2424.00 657.00	−0.175	0.861
*COL4A2*	Negative (61) Positive (17)	38.64 42.59	2357.00 724.00	−0.635	0.525
*MDM2*	Negative (61) Positive (17)	40.05 37.53	2443.00 638.00	−0.405	0.685
*MMP10*	Negative (61) Positive (17)	37.67 46.06	2298.00 783.00	−1.349	0.177
*MMP11*	Negative (61) Positive (17)	37.74 45.82	2302.00 779.00	−1.301	0.193
*MMP13*	Negative (61) Positive (17)	39.82 38.35	2429.00 652.00	−0.236	0.813
*MMP2*	Negative (61) Positive (17)	38.70 42.35	2361.00 720.00	−0.587	0.557
*MMP3*	Negative (61) Positive (17)	38.56 42.88	2352.00 729.00	−0.696	0.486
*MMP7*	Negative (61) Positive (17)	39.67 38.88	2420.00 661.00	−0.127	0.899
*MMP9*	Negative (61) Positive (17)	39.16 40.71	2389.00 692.00	−0.248	0.804
*TGFB1*	Negative (61) Positive (17)	38.95 41.47	2376.00 705.00	−0.405	0.685
*TIMP2*	Negative (61) Positive (17)	37.84 45.47	2308.00 773.00	−1.228	0.219
*TIMP3*	Negative (61) Positive (17)	37.75 45.76	2303.00 778.00	−1.289	0.197
*TIMP4*	Negative (61) Positive (17)	39.08 41.00	2384.00 697.00	−0.309	0.758
*VEGFA*	Negative (61) Positive (17)	40.07 37.47	2444.00 637.00	−0.418	0.676
*NOTCH1*	Negative (61) Positive (17)	40.95 34.29	2498.00 583.00	−1.071	0.284
*OCLN*	Negative (61) Positive (17)	39.38 39.94	2402.00 679.00	−0.091	0.928
*SNAI1*	Negative (61) Positive (17)	37.30 47.41	2275.00 806.00	−1.628	0.104
*SNAI2*	Negative (61) Positive (17)	39.49 39.53	2409.00 672.00	−0.006	0.995
*SNAI3*	Negative (61) Positive (17)	39.72 38.71	2423.00 658.00	−0.163	0.870
*TGFB2*	Negative (61) Positive (17)	39.92 38.00	2435.00 646.00	−0.309	0.758
*TGFB3*	Negative (61) Positive (17)	39.51 39.47	2410.00 671.00	−0.006	0.995
*TIMP1*	Negative (61) Positive (17)	38.46 43.24	2346.00 735.00	−0.769	0.442
*TWIST1*	Negative (61) Positive (17)	39.02 41.24	2380.00 701.00	−0.357	0.721
*VIM*	Negative (61) Positive (17)	39.61 39.12	2416.00 665.00	−0.079	0.937
*ZEB1*	Negative (61) Positive (17)	39.18 40.65	2390.00 691.00	−0.236	0.813
*ZEB2*	Negative (61) Positive (17)	38.92 41.59	2374.00 707.00	−0.430	0.667
*VCAN*	Negative (61) Positive (17)	39.08 41.00	2384.00 697.00	−0.309	0.758
*ADAMTS1*	Negative (61) Positive (17)	39.46 39.65	2407.00 674.00	−0.030	0.976
*ADAMTS13*	Negative (61) Positive (17)	42.08 30.24	2567.00 514.00	−1.906	0.057
*ADAMTS8*	Negative (61) Positive (17)	42.30 29.47	2580.00 501.00	−2.064	0.039
*LOXL2*	Negative (61) Positive (17)	38.13 44.41	2326.00 755.00	−1.011	0.312
*ECM1*	Negative (61) Positive (17)	37.93 45.12	2314.00 767.00	−1.156	0.248
*MMP8*	Negative (61) Positive (17)	38.31 43.76	2337.00 744.00	−0.879	0.380
*VCAM1*	Negative (61) Positive (17)	39.95 37.88	2437.00 644.00	−0.333	0.739
*VTN*	Negative (61) Positive (17)	38.78 42.09	2365.50 715.50	−0.533	0.594
*HIF1A*	Negative (61) Positive (17)	38.51 43.06	2349.00 732.00	−0.732	0.464
*VEGFB*	Negative (61) Positive (17)	39.08 41.00	2384.00 697.00	−0.309	0.758
*VEGFC*	Negative (61) Positive (17)	39.16 40.71	2389.00 692.00	−0.248	0.804

**Table 2 ijms-26-06261-t002:** Top common binding compounds of the initial 23 for ADAMTS8 and its co-dependent genes.

Protein	Ligand Binding Affinity (kcal/mol)			Pan-Target Ligands
	Low (≤6.9)	Medium (7.0–8.9)	High (≥9.0)	
ADAMTS8		Aflatoxin B2, Fulvestrant	Benzo[a]pyrene, Cyanoginosin LR, Doxorubicin	Benzo[a]pyrene, Cyanoginosin LR, Doxorubicin
DNAL4	Benzo[a]pyrene, Doxorubicin, Fulvestrant, Progesterone	Cyanoginosin LR		
EVI2B	Doxorubicin, Estradiol, Progesterone	Benzo[a]pyrene, Cyanoginosin LR		
PPP1R35	Aflatoxin B2, Estradiol	Benzo[a]pyrene, Doxorubicin	Cyanoginosin LR	
PTGR3			Aflatoxin, Benzo[a]pyrene, Cyanoginosin LR, Doxorubicin, Fulvestrant	
RPL21	Benzo[a]pyrene, Doxorubicin, Estradiol, Progesterone	Cyanoginosin LR		
SOX4	Aflatoxin B2, Progesterone	Benzo[a]pyrene, CyanoginosinLR, Doxorubicin		
ZNF3		Doxorubicin, Estradiol, Fulvestrant	Benzo[a]pyrene, Cyanoginosin LR	

**Table 3 ijms-26-06261-t003:** Top common binding compounds of the 9661 similar to ADAMTS8 and its co-dependent genes.

Protein	Binding Affinity (kcal/mol)			Pan-Target Ligands
	Low (≤6.9)	Medium (7.0–8.9)	High (≥9.0)	
ADAMTS8			Dibenzo(a,e)pyrene (9126), 148413, 169380, 626153, 12588587, 14274984, 15001195, 101392782, 129716757, 129805998	Dibenzo(a,e)pyrene (9126)
DNAL4		Dibenzo(a,e)pyrene (9126), 12588587, 14274984, 15001195, 56664790, 101023804, 101392782, 102060712, 129628257, 129716757		
EVI2B		Dibenzo(a,e)pyrene (9126), 159823, 160349, 187315, 14274984, 15001195, 23617881, 129628257, 129716757, 129762518		
PPP1R35		Dibenzo(a,e)pyrene (9126), 153936, 12588587, 14274984, 23617881, 71452697, 102224786, 129628257, 129853608	23621448	
PTGR3			Dibenzo(a,e)pyrene (9126), 42890, 186437, 10099105, 12588587, 13553135, 23617881, 129628257, 129716757, 129805998	
RPL21		Dibenzo(a,e)pyrene (9126), 12588587, 14274984, 23617881, 23621448, 70695534, 90678391, 101023800, 101346118, 129628257		
SOX4		148413, 12588587, 14274984, 129716757, 129853608	Dibenzo(a,e)pyrene (9126), 23617881, 23621448, 101392782, 129628257	
ZNF3			Dibenzo(a,e)pyrene (9126), 160249, 169380, 12588587, 142749984, 23617881, 23621448, 101392782, 129701154, 129716757	

## Data Availability

The original contributions presented in this study are included in the article/Appendix A. Further inquiries can be directed to the corresponding author.

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
