# Peer review of "Identifying Molecular Modulators of the Vascular Invasion in Rectal Carcinoma: Role of ADAMTS8 and Its Co-Dependent Genes"

_ijms, 2025, doi:10.3390/ijms26136261_

Round 1
Reviewer 1 Report
Comments and Suggestions for Authors
-
- The study is primarily computational, and experimental validation of ADAMTS8’s role is currently lacking. Do the authors plan to conduct in vitro or in vivo studies to confirm ADAMTS8’s role in modulating VI?
- Can overexpression of ADAMTS8 reverse the VI phenotype in relevant cell line models?
- Some of the reported correlations appear to be weak. Correlation coefficients in the range of r = 0.3–0.5 are generally considered moderate rather than strong. Could the authors clarify whether such correlations are biologically meaningful in the context of the study?
- Please clarify the threshold used for identifying gene interactions. What criteria were applied to define significance or relevance?
- The authors state: “We identified four high-affinity compounds: cyanoginosin LR, doxorubicin, benzo[a]pyrene, and dibenzo[a,e]pyrene.”
Are these compounds being proposed as potential therapeutic modulators? If so, this raises concerns regarding their safety and feasibility, particularly due to known toxicities. Could the authors address these concerns?
I believe the manuscript requires additional experiments to support the computational analysis results in order to meet the standards of IJMS.
Author Response
Dear Reviewer 1,
we would like to sincerely thank you for the useful and constructive comments that surely led to improvement of our manuscript. We respond to all your queries and suggestions and hopefully met the expectations. The revised text according to your instructions has been marked red.

Reviewer 2 Report
Comments and Suggestions for Authors
In the manuscript “Identifying Molecular Modulators of The Vascular Invasion in Rectal Carcinoma: Role of ADAMTS8 And Its Co-Dependent Genes” the authors show gene pathways and chemical agents most likely associated with VI in rectal carcinoma. This manuscript is interesting and has great potential, however there are some methodological issues that I state bellow:
- In materials and methods section when describing STRING analysis please provide more data about the query (like input proteins, minimum required interaction score, 1st and 2nd shell number of interactors, etc…)
- In the materials and methods section, like in the previous case, when describing The DepMap portal website analysis, please add more data about the query and analysis. Actually, please rewrite section “4.4. Enrichment analysis for vascular invasion-related gene product” describing in detail your analyses as it does not provide enough information for the methodology section.
- Figures 2 and 5 show correlation data and are strongly commented on in discussion section. Please note that your r-values indicate no correlation. Usually 0.7 (-0.7) and above is considered a valid correlation. In your case maybe r=0.501 could be considered as an inclination toward correlation but not an actual correlation. P-value <0.5 only shows the strength of r-value and makes no difference for the conclusion if r-value doesn’t confirm correlation. Please comment. Which r-value did the authors consider as a correlation?
- Please improve the figure images as they are not very neat.
Author Response
Dear Reviewer 2,
we would like to sincere thank you for the useful and constructive comments that surely led to improvement of our manuscript. We respond to all your queries and suggestions and hopefully met the expectations. The revised text according to your instructions has been marked red.

Reviewer 3 Report
Comments and Suggestions for Authors
This manuscript presents a bioinformatics-driven investigation into the molecular underpinnings of vascular invasion (VI) in rectal carcinoma (RC), focusing on the role of ADAMTS8 and its co-dependent genes. Using transcriptomic and methylation data from TCGA-COADREAD (filtered for rectal cancer), and supported by interaction and docking analyses, the authors identify several candidate modulators of VI, notably ADAMTS8, and propose chemical compounds with potential regulatory roles. The study is timely and relevant, considering the poor prognosis associated with VI in RC and the lack of robust biomarkers or targeted therapies.
Overall, the work is well-conceived and executed, although there are key areas where the manuscript could be improved to strengthen its scientific rigor and clarity.
Major Comments:
1. Cohort Size and Statistical Power:
The analysis was performed on a relatively small subset (n=78), with only 17 VI-positive cases. While the authors acknowledge this, more discussion is warranted on the limitations this imposes on statistical power and generalizability.
It would also be helpful to clarify if the data filtering for rectal carcinoma preserved the clinical annotation quality (e.g., VI status determined uniformly).
2. Biological Interpretation of ADAMTS8 Findings:
The finding of lower ADAMTS8 expression being associated with VI is potentially important. However, it is paradoxical that ADAMTS8 showed no interaction in the STRING protein-protein interaction network. This discrepancy between co-expression and lack of network integration warrants deeper interpretation or caution in drawing strong functional conclusions.
Consider integrating more literature context on the proposed tumor-suppressive and anti-angiogenic functions of ADAMTS8 in other cancer types.
3. Co-dependent Gene Analysis:
The selection of the seven co-dependent genes via DepMap is appropriate, but the functional rationale for some genes (e.g., DNAL4, PPP1R35) in the context of RC or VI is unclear. Some gene functions remain obscure and should be briefly described or contextualized better.
4. Molecular Docking:
The docking approach is ambitious, screening >9,000 compounds. However, experimental validation is missing. While the authors do acknowledge this limitation, they should be more cautious in suggesting translational relevance without in vitro or in vivo support.
The use of compounds such as benzo[a]pyrene and dibenzo(a,e)pyrene—known carcinogens—should be carefully framed to avoid misinterpretation that they may be therapeutically useful without toxicity assessment.
5. The supplementary tables are detailed and essential. Consider moving key parts (e.g., Table 1S with gene associations to VI) into the main text or summarizing them in a figure for clarity.
Minor Comments:
1. Use consistent terminology for VI (e.g., vascular invasion vs. venous invasion vs. EMVI). Clarify early that VI includes both intramural and extramural types if this is the case.
2. Figures 1–5 are informative but could benefit from clearer labels and legends. Highlighting significant p-values in figure annotations would improve readability.
3. Ensure that all in-text citations (e.g., [10–31]) are correctly matched with the reference list and accessible for peer reviewers.
Author Response
Dear Reviewer 3,
we would like to sincere thank you for the useful and constructive comments that surely led to improvement of our manuscript. We respond to all your queries and suggestions and hopefully met the expectations. The revised text according to your instructions has been marked red.

Round 2
Reviewer 1 Report
Comments and Suggestions for Authors
-
Author Response
Dear Reviewers,
thank you very much for taking your time to review our manuscript and provide useful feedback.
The response and action are made in manuscript body text accordingly.
Best regards,
Authors

Reviewer 2 Report
Comments and Suggestions for Authors
I thank the authors for the effort to address all my concerns. I think the manuscript has improved and could be ready for publishing after one final correction.
Please note that the r-value shows the strength and direction (positive/negative) of the relationship between two variables. Less than 0.5 means low correlation, while less than 0.3 means very low or even negligible correlation. On the other hand, p-value <0.5 only says that the observed correlation is statistically significant and not due to random chance. After reading your response I understand that you still consider them important even though your results show a real (though weak or very weak) relationship between the variables. Please make sure to correct all mentions of the correlation throughout the manuscript and make sure to state “weak or very weak statistically significant correlation”, especially if it says “strong correlation” since it could be misleading and definitively not correct way of describing your results.
Author Response

(The authors gave the same response as above.)

Reviewer 3 Report
Comments and Suggestions for Authors
The authors have thoroughly addressed the concerns raised in the previous round of review.
I am satisfied with the revisions and agree that the manuscript is suitable for publication in its current form.
Author Response

(The authors gave the same response as above.)
